# Dissecting the Seed Maturation and Germination Processes in the Non-Orthodox *Quercus ilex* Species Based on Protein Signatures as Revealed by 2-DE Coupled to MALDI-TOF/TOF Proteomics Strategy

**DOI:** 10.3390/ijms21144870

**Published:** 2020-07-09

**Authors:** Besma Sghaier-Hammami, Sofiene B.M. Hammami, Narjes Baazaoui, Consuelo Gómez-Díaz, Jesús V. Jorrín-Novo

**Affiliations:** 1Centre de Biotechnologie de Borj-Cédria, Laboratoire des Plantes Extrêmophiles, BP 901, Hammam-Lif 2050, Tunisia; 2Agroforestry and Plant Biochemistry, Proteomics and Systems Biology, Department of Biochemistry and Molecular Biology, University of Cordoba, UCO-CeiA3, 14014 Cordoba, Spain; 3Institut National Agronomique de Tunisie (INAT), Laboratoire LR13AGR01, Université de Carthage, 43 Avenue Charles Nicolle, Tunis 1082, Tunisia; sofiene.hammami@inat.u-carthage.tn; 4King Khalid University, Abha 61421, Saudia Arabia; baazaouinarjes@gmail.com; 5Unidad de Proteómica, Servicio Central de Apoyo a la Investigación (SCAI), Universidad de Córdoba, 14014 Córdoba, Spain; bc2godic@uco.es

**Keywords:** *Quercus ilex*, acorns, non-orthodox seeds, seed maturation, seed germination, proteomics, embryo axis, cotyledon, inviable seeds

## Abstract

Unlike orthodox species, seed recalcitrance is poorly understood, especially at the molecular level. In this regard, seed maturation and germination were studied in the non-orthodox *Quercus ilex* by using a proteomics strategy based on two-dimensional gel electrophoresis coupled to matrix-assisted laser desorption ionization/time of flight (2-DE-MALDI-TOF).Cotyledons and embryo/radicle were sampled at different developmental stages, including early (M1–M3), middle (M4–M7), and late (M8–M9) seed maturation, and early (G1–G3) and late (G4–G5) germination. Samples corresponding to non-germinating, inviable, seeds were also included. Protein extracts were subjected to 2-dimensional gel electrophoresis (2-DE) and changes in the protein profiles were analyzed. Identified variable proteins were grouped according to their function, being the energy, carbohydrate, lipid, and amino acid metabolisms, together with protein fate, redox homeostasis, and response to stress are the most represented groups. Beyond the visual aspect, morphometry, weight, and water content, each stage had a specific protein signature. Clear tendencies for the different protein groups throughout the maturation and germination stages were observed for, respectively, cotyledon and the embryo axis. Proteins related to metabolism, translation, legumins, proteases, proteasome, and those stress related were less abundant in non-germinating seeds, it related to the loss of viability. Cotyledons were enriched with reserve proteins and protein-degrading enzymes, while the embryo axis was enriched with proteins of cell defense and rescue, including heat-shock proteins (HSPs) and antioxidants. The peaks of enzyme proteins occurred at the middle stages (M6–M7) in cotyledons and at late ones (M8–M9) in the embryo axis. Unlike orthodox seeds, proteins associated with glycolysis, tricarboxylic acid cycle, carbohydrate, amino acid and lipid metabolism are present at high levels in the mature seed and were maintained throughout the germination stages. The lack of desiccation tolerance in *Q. ilex* seeds may be associated with the repression of some genes, late embryogenesis abundant proteins being one of the candidates.

## 1. Introduction

Protein signatures, experimentally afforded by proteomics, determine the phenotype, characteristics and properties of biological systems and processes. We have applied this principle to study seed development and germination in the non-orthodox *Quercus ilex,* the most representative and emblematic forest tree species of the Mediterranean forest and of the agrosilvopastoral ecosystem “dehesa” [1]. Despite being a priority, reforestation and breeding programs with this species are hindered by its long lifespan, lack of natural regeneration, and its biological properties. These latter, impede a viable and feasible germplasm conservation and propagation, which is also linked to the lack of knowledge of its biology, especially at the molecular level [1].

Seed formation and germination are unique characteristics of spermatophytes which support propagation in mostplants, and they are crucial not only for plant development but for human food supply, the conservation of genetic resources, and breeding programs [2].

Depending on the seed’s characteristics, properties, maturation, and germination, plant species are classified as recalcitrant (non-orthodox), intermediate or orthodox [3]. The main difference is their sensitivity to water loss and desiccation. Recalcitrant seeds are those that lose viability when dried to a water content of less than 12%, and that lack a dehydration stage during seed ripening [4]. Intermediate and recalcitrant seeds are unable to survive drying and chilling as they rapidly lose their germination capacity and viability during storage [5]. Therefore, for these species, long-term germplasm conservation using conventional seed storage methods is not possible and the development of an alternative strategy remains a real challenge.

The genetic, genomic, and molecular bases of seed development and germination have been extensively studied and acquired in the case of orthodox seeds but are poorly understood in the recalcitrant. Seed maturation and germination, and the transition from one stage to the other, implies complex physiological and biochemical processes. They are induced by environmental factors and under the control of phytohormones and other signaling molecules. They were further mediated by genetic and epigenetic gene expression programs [6,7,8,9]. This results in stage-specific transcriptomic, proteomic and metabolomic profiles. They are characterized by structural and functional signatures including operating metabolic pathways, reactive oxygen species (ROS) production, synthesis, accumulation, and mobilization of reserve proteins, and the production of antioxidant and other stress-related proteins, among others [10,11]. Recalcitrant seeds are able to germinate immediately after shedding, thus implying an active metabolism in the mature stage, a property that is absent in quiescent orthodox seeds [12].

As a continuation of our previous work published on *Q. ilex* acorns and seed germination [13,14,15], we have undertaken a proteomic analysis of seed development and germination. Proteomics has been previously employed in the study of seeds [16], and a number of papers that have focused on germination and viability in Quercus spp. have been previously published [17,18]. The novelty of the present work was the study of the maturation and germination processes as a continuum, and the independent analysis of the cotyledon and the embryo axis throughout 14 stages, nine for maturation and five for germination. From the stage-specific protein profiles visualized, the molecular basis of seed development in this non-orthodox *Q. ilex* species can be hypothesized. This and similar studies would help in establishing clonal propagation methods by means of somatic embryogenesis through the monitoring and molecular comparison of somatic, mostly inviable, and zygotic, mostly viable, embryos [19], as well as the establishment of ex situ germplasm cryoconservation protocols [18].

## 2. Results

### 2.1. Acorn Maturation and Germination Stages

The present study is a continuation of our previous work [13], in which the protein profiles of the Holm oak seed cotyledon and the embryo axis at the mature, harvesting, stage were compared. In this paper, the changes in protein profiles were analyzed throughout different acorn developmental (middle, M4–M6, and late, M8–M9 stages, with M9 corresponding to the mature acorn) and germination (early, G1–G3, and late, G4–G5 stages) phases (Appendix A).

Apart from the visual aspect, morphometry, and weight (Table 1), the different stages were characterized by tissue water content (Figure 1).

In cotyledons, the water content decreased from 80% at M4 to 35% at the mature, M9, stage (Figure 1A). Upon imbibition, water content increased at G1 up to 76%, then decreasing again to 35% at G5. The water content in the embryo axis remained nearly constant, 75% throughout the maturation stages and a decrease to 53% was observed only at M9. Upon imbibition, water content increased to 89% and remained above 62% at the germination stages analysis (Figure 1A).

### 2.2. Protein Extraction, 1- and 2-D Gel Electrophoresis

Proteins were extracted by a trichloroacetic acid-acetone/Phenol/chloroform method and quantified by Bradford (Figure 1B). During maturation, the protein content was much higher, in relative terms (per g of tissue dry weight), in the embryo axis than in the cotyledon. In the embryo axis, an initial increase, with maxima at M6 and M7 (around 15 g/g DW) was followed by an ulterior decline to values of below 2 g/g DW at M9. During all germination stages, the amount of proteins remained constant, with similar values to M9, mature acorns. For the cotyledon, the protein content remained nearly constant at all stages from M4 to G5, with values of around 3 g/g DW. For the non-germinating (NG) acorns, cotyledon and the embryo axis presented a similar protein content corresponding to around 4 g/g DW.

The protein extracts from seeds at different developmental stages (from M4 to G5) and tissues (embryo axis, cotyledon) were analyzed by 1-D electrophoresis to discriminate the most important stages on which changes occurred during seeds development (Figure 2A, Appendix A, and Appendix A).

The number of bands resolved was 38–50 (cotyledon), and 23–58 (the embryo axis). The highest number of bands corresponded to the M9 stage in cotyledon, whereas in the embryo axis they corresponded to M6–M9 (Appendix A). During maturation stages, thirty-six out of the 50 different bands were variable in cotyledon, while 32 out of 58 were variable in the embryo axis. During germination, the number of bands decreased in the embryo axis /radical; however, it remained constant in cotyledon. Twenty out of 53 protein bands were qualitatively variable in cotyledon, while 15 out of 52 were variable in the embryo axis. The NG acorns presented more protein bands in embryo axis than in cotyledon (Appendix A).

Based on the results obtained by 1-DE, a 2-DE analysis (Figure 2B, Appendix A) was then performed on the last four development stages (M6 to M9, representing middle and late stages) and on G3 and G5 germination ones (representing early and late stages). It is there where most of the changes in protein profiles were observed. Some variability in the 2-DE pattern among replicates was observed and, consequently, the consistent spots, i.e., those present in all three replicates (representing about 70% of the total), were considered in the statistical analysis (Appendix A). During maturation, the number of consistent spots ranged between 446–506 (cotyledon), and 326–470 (the embryo axis), with the maximum value corresponding to M6–M8 for cotyledon, and M9 for the embryo axis. The number of spots decreased during germination compared to the M9 stage both in cotyledon and the embryo axis (Table 2).

By comparing the different maturation and germination stages in each part of the seed tissue, a total of 558 and 591 spots showed significant changes (±1.5-fold, *p* < 0.05, Table 2) in abundance in cotyledon and the embryo axis, respectively. The NG acorns presented more spots in embryo axis (388) than in cotyledon (234) (Table 2).

Some of the spots (10% of identified proteins) showed qualitative differences (presence/absence) between samples and can be proposed as markers of viability, maturation and germination stages, and tissues (cotyledons and embryo). The list of those markers included:

(i) Markers of viability, which were absent in non-germinating seeds, spot 7503 (with the identified protein being an Alcohol dehydrogenase), spot 6425 (lactate/malate dehydrogenase), spots 308, 1312, 2313 and 2315 (with the identified protein being an Enoyl-ACP reductase precursor) and spot 8630 (with the identified protein being a dehydrin protein).

(ii) Markers of developmental stages, which were abundant only in germination, spots 5123 and 6102 (with the identified protein being an Osmotin), so they were chosen as biomarkers for germination. However, spots 2406 and 2603 (with the identified protein as ATP synthase alpha/beta family protein) were only abundant in late maturation, so they were chosen as biomarkers for this phase. The spots 4834, 5819, 6801 and 2406 (with the identified protein being a Transketolase) were highly abundant at middle maturation stages, and spot 8630 (with the identified protein being dehydrin) was highly abundant only at late maturation stages.

(iii) Markers of storage tissues, which were abundant only in cotyledon, spots 8705, 6125, 7232 and 7307 (with the identified protein being a Legumin precursor), spots 17, 1013, 2005, 3009, 6002, 8006, 8008 and 8110 (with the identified protein being a RmlC-like cupins superfamily protein) and spots 5212, 4220 and 4218 (with the identified protein being a 11S globulin isoform 3).

### 2.3. Statistical Analysis of the Data

2-DE spot intensity data were subjected to multivariant, Venn (Figure 3) and principal component analysis (PCA) (Figure 4) during maturation and germination stages, and for seed parts, the cotyledons and the embryo axis.

During cotyledon’s development, the different maturation stages (M6–M9) had 407 common protein spots, while the germination stages (G3–G5) shared only few common proteins (145) (Figure 3). Contrarily, during the embryo axis development, the maturation (M6–M9) and germination (G1–G3) stages shared a higher number of proteins, 316 and 405, respectively (Figure 3). The embryo axis of the non-germinating acorn (ENG) and that of EG3 have 74 common spots, but the cotyledons of NG and CG3 only have 20.

In the first PCA performed, comparing stages, the first two components accounted for 57% variability (35% for PC1, and 22% for PC2), and 62% variability (46% for PC1, and 17% for PC2) for the cotyledon and the embryo axis, respectively, with the first seven components accounting for 99.99% of the biological variability of the dataset in all the developmental stages (Appendix A).

All cotyledon and embryo axis maturation stages were clearly separated from the germination stages by PC1 (Figure 4A,B). PC2 discriminated, on one hand, cotyledons from non-germinating and germinating ones (CG3 and CG5), and on the other hand, early maturation (CM6–CM8) from late (CM9) stages (Figure 4A). For the embryo axis, PC2 separated EM6–EM7 from EM8–EM9. However, at the germination stage, PC2 grouped the embryo axis of non-germinating acorns (ENG) with EG3 and separated it from EG5 (Figure 4B). Based on PC1, 228 and 41 spots decreased in intensity from maturation to germination in the cotyledon and the embryo axis, respectively. Conversely, six spots in the cotyledon and 119 in the embryo axis increased in intensity from maturation to germination (Appendix A).

With regard to the second two PCA analyses, when comparing seed parts (Figure 4C,D), the first two components accounted for 68% variability (50% for PC1, and 18% for PC2), and 57% variability (35% for PC1, and 22% for PC2) for the maturation and germination stages, respectively. The seven principal components accounted for 99.99% of the biological variability of the dataset in all developmental stages (Appendix A). It is clear that PC1 discriminated the maturation of the embryo axis from that of cotyledon, and similarly for the germination stages (Figure 4C,D). The PC2 discriminated germination and maturation stages similarly to what was observed in the previous analyses (i) and (ii). Based on PC1, 185 spots were more abundant during maturation in cotyledon than in the embryo axis, and, inversely, 162 spots were more abundant during germination in embryo axis than in cotyledon (Appendix A).

Similar results in differences and distances between samples were obtained when performing a hierarchical clustering analysis (Appendix A).

### 2.4. Protein Identification, Functional Classification of the Differentially Abundant Proteins, and Changes in Abundance Throughout the Maturation and Germination Stages

The selection of spots that could be used as biomarkers for each developmental stage would be of great interest. The loading of each particular spot to PC 1, 2, and 3 was determined from the factor matrix generated during the PCA. A total of 160 (for cotyledon) and 57 (for embryo axis) spots showed the highest correlation (above |0.9|) with each PC determined (Appendix A). These spots can be used to differentiate the stages studied and, once identified, to improve the explanation of the variability contained in those groups.

Variable spots were excised from the gel and subjected to matrix-assisted laser desorption ionization/time of flight (MALDI-TOF) analysis. For identification, the *Quercus ilex* homemade and the National Center for Biotechnology Information (NCBI) databases were used. Only proteins with a high confidence in the (peptide mass fingerprinting) PMF matches, based on a score of over 71, were considered significant (*p* < 0.05). Out of the total variable spots, 178 (cotyledon) and 143 (embryo) were successfully identified (Appendix A). In general, for the proteins identified, *Mr* values were overestimated under the electrophoresis conditions employed, or, in some cases, as reserve proteins, that may be the result of protein aggregation. On the other hand, the experimental and theoretical p*I* were very close. Some of the proteins identified were present in spots having different *Mr* and p*I* and they corresponded to isoforms or proteoforms as a result of post-translational modifications (PTMs). That is the case of phosphoglycerate mutase, enolase, and NAD(P)-linked oxidoreductase superfamily protein in embryos, and reserve proteins, pyruvate kinase, pyruvate dehydrogenase, in cotyledons (Appendix A).

The 178 proteins identified in the cotyledon could be classified into five functional and 10 sub-functional groups, based on their putative function according to the Kyoto Encyclopedia of Genes and Genomes (KEGG). The most abundant categories were associated essentially with energy metabolism (47%), storage proteins (13%), cell defense and rescue (11%) and protein biosynthesis and destination (10%). The 143 proteins identified in the embryo axis could be classified into seven functional and eight sub-functional groups. The most abundant categories were associated with energy metabolism (21%), cell defense and rescue (31%) and protein biosynthesis and destination (12%) (Appendix A).

To understand the changes in the proteome of the different tissue parts of non-orthodox seeds during development and germination, the protein abundance of cotyledon and the embryo axis were compared using the Genesis software package. In addition, the different expression dynamics of each protein identified was represented by a heat map (Figure 5A,B). The trends of those proteins in the two different tissue parts during all the developmental stages are summarized in Figure 6.

As a general rule, in cotyledon, a similar tendency was observed for most of the groups. Protein abundance was maximum at M6, and then decreased at M9 (mature stage) to low values, which were maintained at the germination stages. The exception was in the reserve proteins, whose abundance increased from M6, presenting maximum values at M8 and M9. Some of them were mobilized and hence decreased in abundance at M9 and germination stages. More tendencies were observed in embryo proteins. Some of the groups, such as those of the energetic and carbohydrate metabolism, increased from M6 or M8, having maximum values at the germination stages. Protein abundance in the reserve proteins, and stress responses, including the detoxification of reactive oxygen species, increased from M6 to M8 or M9, then decreased to low values throughout the germination stages for those with maximum values at M8, or remaining at the M9 high levels. For other groups (amino acid metabolism, lipid metabolism, protein biosynthesis and proteolysis) no specific trends were obtained.

## 3. Discussion

As a continuation of the previously published work on proteomics of mature acorns from the non-orthodox *Quercus ilex* tree [13], the protein signatures of cotyledons and the embryo axis throughout seed development and germination phases have been established from data obtained using a 2-DE coupled to MALDI-TOF/TOF MS proteomics strategy.

The protein profiles at six maturation (M4–M9) and five germination (G1–G5) stages, with M9 corresponding to the mature acorns at shedding, were analyzed. Non-germinating seeds after a two-week period upon imbibition were also included in the study. Acorns started to be harvested once they were clearly visible on the tree (M1). Then, the different stages (M1–M9) were chosen based on periodical surveys and visual differences (Appendix A), resulting in differences in their morphometry (width, length), weight, and water content (Table 1, Figure 1A). Almost identical stages were employed in the transcriptomic profiling of developing cork oak acorns [20].

Water content is crucial for viability in recalcitrant seeds, especially in the embryo axis [4], which is protected from dehydration by the waterproof pericarp, seed coat, and the surrounding cotyledon [21]. The lowest water content values corresponded to the mature shedding stage (around 35 and 55% for the cotyledon and the embryo axis, respectively) (Figure 1A). Upon imbibition, germination starts, and seed tissues become hydrated (around 75 and 85% for cotyledon and the embryo axis, respectively, at G2). Even though the relative water content of seeds was above the percentage of a viable one (around 35%), around 30% of the acorns did not germinate in the two weeks after imbibition. On the other hand, and as synchronized germination does not occur in recalcitrant *Quercus* spp. [22] the stages and parameters corresponded to the most representative fruits on each date.

Although 1-D gel electrophoresis is not very resolutive, it is simple and useful when there are a large number of samples, as in the present work (around 30 samples including stages and replicates). This technique allowed us to discriminate between samples, and to group them [15]. In the present research and based on differences in the 1-D protein band profiles (Figure 2A, Appendix A), the last four developmental stages M6 to M9, corresponding to filling and maturation. This was deduced from the increase in weight and decrease in water content. The two germination stags, G3 (early) and G5 (late), were selected for 2-DE-MS analysis.

2-D gel electrophoresis resulted in 300–500 consistent spots resolved in the 5–8 pH range, with maximum values at M6–M8 in cotyledons, and M9 in the embryo axis (Figure 2B, Appendix A and Table 2). This is in good agreement with the total protein amount in the extracts as determined by the Bradford assay (Figure 1B). Similar comparative values in protein abundance between stages have been shown in different seed proteomics studies [23]. Storage proteins (legumin, 11S globulin, RmlC-like cupins superfamily protein) are those most abundant in the seed [15,24], and, according to the data presented, they are accumulated earlier in cotyledons than in the embryo axis, although the relative abundance is much higher in the former. In a related paper, it has been shown that storage proteins, mostly 7S globulins, are those most commonly accumulated in embryo and endosperm, although with higher values in endosperm at the mature drying stage [25]. The synthesis and accumulation pattern of reserve proteins seems to be quite similar in different plant species, independently of their botanical group (monocot, dicot) and seed characteristic (orthodox or recalcitrant), although there must be differences in the accumulation kinetics for individual seeds [26]. As it is well known for different plant systems [27], the degradation and mobilization of reserve proteins occurs at early germination stages, as revealed by a significant decrease in the number of resolved spots in both cotyledon and the embryo axis (Table 2).

In non-germinating seeds, and according to the protein content values (3.2 and 3.3 g/g DW in the embryo axis and cotyledon, respectively), the 1-D (40 and 46 resolved bands for, respectively, cotyledon and embryo axis) and 2-D (234 and 388 resolved spots for, respectively, cotyledon and embryo axis) profiles, we can speculate that the loss of viability may be associated with the inability to synthesize and accumulate some reserve proteins among others [28].

The 2-DE profile was stage specific, with 558 (cotyledon) and 591 (the embryo axis) variable spots in all the samples (Table 2). In the PCA analysis, PC1 clearly separated the maturation and germination phases, including non-germinating acorns and tissues, and PC2 stages (Figure 4), together explained 56–67% of the variability. From PC2 data, we can clearly differentiate CM6-CM8 from the CM9 stages and EM6/EM7 from EM8/EM9 stages. According to the variable spots, the cotyledon and the embryo axis behaved differently. Thus, in the former, in the transition from maturation to germination, 228 spots decreased in abundance and only six increased. In contrast, in the embryo axis, 119 increased and 41 decreased (Appendix A). So, most metabolic activity should occur in cotyledon at early stages and in the embryo axis at the late ones. Thus, 185 spots were more abundant in cotyledon than in embryo axis at maturation, while 162 were more abundant in the embryo than in the cotyledon at germination. The embryo axis had a large number of common spot proteins in the late maturation and early germination stages (Figure 3), pointing to, unlike orthodox seeds, a continuum between maturation and germination.

Out of the total variable spots, 211 (cotyledon) and 181 (the embryo axis) were subjected to MALDI-TOF/TOF MS with 178 and 143, respectively, were identified (Appendix A). Identified proteins were classified into five (embryo) and seven (cotyledon) functional groups, with 10 and 8 pathways represented within the metabolic group. Next, the trends observed for each group and pathway will be discussed.

An important metabolic activity at specific stages of maturation and germination phases does occur according to the protein signatures, and takes place, as a general tendency, earlier in the cotyledon (M6–M7) than in the embryo axis (M8–M9). Thus, maximum protein abundance was observed in cotyledons during the middle seed filling stages (M6–M7) for glycolytic enzymes (e.g., enolase, and phosphoglycerate kinase), which moved during the late stages, M8–M9, in the embryo axis. The coexistence of anaerobic as well as aerobic energy metabolism can be deduced from the presence in the set of variable proteins of those corresponding to enzymes of TCA cycle, oxidative phosphorylation (ATP synthetase and fermentative alcohol dehydrogenase and lactic dehydrogenase). They were found in both tissues with a similar evolution pattern to that of glycolytic enzymes in cotyledon and maximum at late (M9) and middle germination stages in the embryo (EG3).

The NAD(P)H/NAD(P) balance and homeostasis is one of the key features in metabolism. It is mostly determined by a balance between catabolic oxidative reactions and fermentation reactions and to a lesser extent (based on protein abundance data) of the pentose-phosphate pathway. The abundance of pentose-phosphate pathway enzymes (Glucose-6-phosphate dehydrogenase 6, Transketolase) was much greater in cotyledon (maximum at middle M6) than in embryo (maximum at early G3).

As previously discussed, the presence of RubisCO was deduced from data on the large subunit in both seed parts with higher values in cotyledon than in embryo. Maximum values that were found at M7 in cotyledons, and M8–M9, in the embryo, are controversial. This may be because the seeds are photosynthetically active at early development or later on at the seedling stage, which is manifested at the G5 peak in cotyledon. It can be speculated that the presence of RubisCO may contribute to CO_2_ reassimilation after decarboxylation reactions [13,29].

Methyl glyoxal and other reactive and toxic aldehydes are by-products of the glycolytic route and are thus assumed to be accumulated at the maturation and germination stages, which must be detoxified, this taking place through the methyl glyoxal pathway [25]. Enzymes of this pathway were identified in both cotyledon and embryo (Glyoxalase I and II, and lactoyl glutathione lyase) with peaks at M6 (cotyledon) and M9-early germination stages, in the embryo, coincident, in both cases, with a maximum of glycolytic enzymes.

Seed germination and early seedling growth is mostly supported by reserves stored in cotyledons [30]. In the case of Holm oak cotyledon, reserve nutrients are stored as starch and proteins with lipids being the minor components [15]. Seed filling and storage accumulation occur at middle stages just prior to drying and maturation and paradoxically “*Final grain weight showed few or no significant correlations with enzyme activities, sugar levels, or starch content during grain filling, or with starch content at maturity*” [31].

Seed reserve proteins play a vital role as they are the source of amino acids and precursors of other N compounds, they are present mostly in cotyledons and to a lesser extent in the embryo axis. Those of *Q. ilex* mostly belong to the legumin family, also with some representatives of the cupin (two different RmlC-like cupins superfamily protein in cotyledon and embryo), globulin (11S globulin isoform 3 in cotyledons), and glutelin (Glutelin type-B 5 in the embryo) ones [13]. Two and five different types of legumins were identified in the embryo and cotyledon, respectively, being present as precursors mature forms or degradation products. Legumins, cupins, and globulins started to accumulate at middle M6 in both seed parts, with the maximum at late (M8–M9) maturation stages, they decreased at germination. This pattern is characteristic of most seeds, independently of their orthodox or recalcitrant character [32]. The exception was observed for a couple of members of the cupin family in cotyledons that showed a clear peak at germination stages. This observation is in the direction of proposing alternative roles for this family of proteins [33].

The protein biosynthesis and degradation processes are behind protein accumulation. In the present work, and within the set of variable proteins, three in the embryo and four in the cotyledons related to protein biosynthesis have been identified. They are assumed to have a specific role in seed maturation and germination (Appendix A). This is the case of a Glycyl-tRNA synthetase, present in both tissues, whose gene inactivation in *Arabidopsis thaliana* led to plant embryo development arrest [34]. Ribosomal proteins and elongation factors have also been implicated in seed development and dormancy in *N. tabacum* and *A. thaliana* [35]. The evolution pattern of these proteins is similar to that reported for other groups in cotyledons, with the maximum at M6–M8. In the embryo axis, they were present throughout all the stages analyzed with accumulation kinetics and maximum values depending on the protein.

Extensive protein degradation occurred at late seed development and germination, as revealed by the identification of reserve protein degradation products, up to 10 different ones for legumins and cupins. These degradation products are observed at late maturation (M8–M9) and germination stages, in cotyledons, and, to a lesser extent, in the embryo axis. Degradation products of the RubisCO large subunit also appeared in the cotyledon at germination stages.

According to the set of variable proteins identified, protein degradation is carried out either by unspecific proteases or by the proteasome complex [36]. Among the former, a cytosol aminopeptidase family protein, Zinc metalloprotease, and Aspartic proteinase A1, were identified in the embryo axis, and Aspartic proteinase 1, Peptidase M1 family protein, AAA-type ATPase family protein, in cotyledons. Even though most of them are not well characterized, their presence in seeds has been previously reported. Among these proteins, an Aspartic proteinase 1, the COP9 signalosome, and the AAA-type ATPase family protein regulator have been implicated in the dormancy, viability, and germination processes in Arabidopsis seed [37]. While having maximum values at M7 in cotyledons, protease abundance decreased at late maturation 9, having very low levels at germination or almost none in non-germinated seeds. Maximum values in the embryo axis took place at M8, keeping it more or less constant at germination stages, except in non-germinating seeds.

Other pathways are less well represented in the set of variable proteins, thus preventing any discussion on their role and relevance in seed development, but just mentioning them and verifying their presence in other seeds. Those proteins were linked to carbohydrate, amino acids, and lipid metabolism.

Within the carbohydrate metabolism in the embryo axis, the list included enzymes implicated in the synthesis of starch, the most abundant compound in acorns [15] including phosphorylases and Glucose-1-phosphate adenylyl transferase [38]. Other enzymes identified that were previously reported in seeds were UDP-Glycosyl transferase (an activity involved in the glycosylation of different metabolites; [39], beta glucosidases, UDP-glucose pyrophosphorylase 2 (whose gene inactivation impeded seed set in rice); [40], UDP-D-apiose/UDP-D-xylose synthase 2 (related to cell wall formation [41]. The set in cotyledons comprises the sugar isomerase (SIS) family protein [42], the UDP-glucose 6-dehydrogenase family protein [43], glucose-1-phosphate adenylyl transferase, sucrose synthase 2 [44], GDP-D-mannose 3′,5′-epimerase [45]. The evolution in cotyledons was similar to the previous ones, with the maximum at M7–M9 with later a decay/degradation at germination. In the embryo axis, the peak is displaced at M8–9 and the values are kept at germination stages.

Amino acids play a key role in seed central metabolism, that is used for the synthesis of storage proteins and others, catabolic fuel, precursor of vitamins, hormones, and secondary metabolites [46]. Up to four proteins in the embryo and 18 in the cotyledon, corresponding to enzymes of the amino acid metabolism, have been identified. They showed a similar trend to the other groups in cotyledons, with maximum values at M6–M8 and later a decay at low levels at mature and germination stages. The protein abundance values in the embryo were lower than in cotyledon and a general trend was not observed. Some of them, like alanine aminotransferase, have been reported to play a role in seed dormancy [47].

The list of proteins identified was related to: Arg (embryo, arginosuccinate synthase, [48]; cotyledon carbamoyl phosphate synthetase A: spot, [49]; Glx (embryo, glutamine synthase [50]; cotyledon glutamine amidotransferase type 1 family protein: N-acetyl-gamma-glutamyl-phosphate reductase, and glutamate-1-semialdehyde 2,1-aminomutase 1); Cys (embryo cysteine synthase, [51]; ser (cotyledon d-3-phosphoglycerate dehydrogenase [52]; eu, Ile (cotyledon putative, 2-isopropylmalate synthase 1 [53]; ketol-acid reductoisomerase, Phe/Tyr (cotyledon 3-deoxy-D-arabino-heptulosonate 7-phosphate synthase 1: dehydroquinate dehydratase, putative/shikimate dehydrogenase;3-dehydroquinate synthase, putative [54].

A relevant datum of the fatty acid composition in *Q. ilex* seed is that its profile is quite similar to that of olive oil, with oleic acid being the most abundant (around 60%), followed by linoleic and palmitic (around 18%) and finally stearic (around 4%). Up to three different proteins of enzymes implicated in fatty acid metabolism have been identified, including Enoyl-ACP reductase and 3-ketoacyl-acyl carrier protein synthase, in cotyledon and embryo, plus acetyl co-enzyme, a carboxylase biotin carboxylase subunit, only in cotyledon. The evolution of the proteins throughout maturation and germination, in both the embryo and the cotyledon, is similar to the above discussed for other groups.

Desiccation tolerance in orthodox seeds, and seed viability and longevity depend, to a great extent, on activation, on defense and stress-related genes. Thus, it is not surprising that cell defense and rescue was the functional group most represented in the preformed protein analysis, with two main sub-groups within it, corresponding to redox homeostasis and ROS detoxification, and a more heterogeneous group, including HSPs, and glycine-rich RNA-binding, among others.

Heat-shock proteins (HSPs) are ubiquitous and act as molecular chaperones, thus favoring a competent protein folding and preventing protein aggregation, among other roles, with different families classified according to their size. Differently from vegetative tissues, they are constitutively present in seeds, with differences related to the accumulated type, that have been implicated in desiccation tolerance and with some of them having relevant roles [55]. In the present study, HSPs were identified in the cotyledon (70 type, previously reported in vegetative tissues of Quercus spp., [56], and the embryo axis (70, 20, and 17.9)). The relative abundance was similar in both parts, with a maximum at M6–M8 in cotyledons and a later decay/degradation to M9 and germination, and M7–M9 in the embryo axis, with some of them having high values at germination. This pattern is similar to the one reported for most of the seeds studied [7].

Another family of stress-related proteins well represented in *Q. ilex* seeds was the Glycine-rich RNA-binding one, also involved in signaling and development [57]. Based on relative abundance, some major proteoforms showed some characteristics not observed for other stress-related proteins and groups of them. First, they accumulated at earlier stages than M6, and, second, they were present in embryos in a larger amount throughout the M7–M9 and germination stages.

Within the drought stress-related proteins, two more were identified, including Dehydrin 2 (embryo), and Annexin 1 (embryo and cotyledon). Dehydrin genes have been studied in detail in *Quercus robur*, and their presence confirmed in seeds, being implicated in maturation [58]. Dehydrin transcripts were also detected in germinating *Q. ilex* seeds, with transcript abundance being maximum at mature acorn, then decreasing at later germination stages [14]. The expression of annexins at high levels has been reported for a number of seeds [59].

Proteins related to responses to biotic stresses and plant immunity are linked to specific developmental stages, as is the case for seed maturation and germination, the key stages in the plant biological cycle [7]. This is without discarding the fact that the presence of an endogenous microflora also keeps defense genes activated. In this regard, unpublished metabolomic data by the authors’ laboratory, have revealed the existence, in cotyledons, of fungal and bacteria-derived metabolites. The pathogenesis-related proteins identified in embryo included osmotin 34 [60] and Major allergen Pruar 1 [61].

Seed viability and longevity relies on antioxidants and ROS scavengers [7]. ROS species are produced as a consequence of metabolic reactions in which participate oxygen, and, because of its reactivity, its concentration should be kept below toxicity levels. However, they also act as signaling molecules implicated in development and stress response gene expression regulation. It has been well documented how these species are produced during the seed maturation and germination phases and some hypotheses on their role in signaling and protein redox states have been proposed [62].

The set of antioxidant and redox-related enzymes identified include a Cu/Zn superoxide dismutase, ascorbate and dehydroascorbate peroxidase, Thioredoxins, NAD(P) linked oxidoreductase (the embryo axis and cotyledons), and glutathione-S-transferase (only in embryo). All of them have been previously reported in seeds, being related to seed development and longevity [62]. They have a different evolution pattern; thus, GST appeared mostly in embryo at M7–M9 and at germination stages. Thioredoxins proteoforms had maxima in cotyledons at M7 (101), M9 (3103), later decaying at germination stages, while in embryo they corresponded to G3. NAD(P)-linked oxidoreductase had the standard pattern observed in cotyledons and embryo, maximum at M7–M8 and ulterior decay for the former, and M8–M9 for the latter. Finally, the SOD pattern was similar for cotyledon, and had more or less high constant levels throughout all the stages.

Taking all data together enables us to discuss the differences between viable, germinating, and non-viable, non-germinating seeds, cotyledon and the embryo axis, and orthodox and recalcitrant seeds.

The simple visualization of the gels showed that the number of spots was much lower in non-viable, non-germinating seeds than in viable, germinating ones at M9, so the loss of viability may be associated with the inability to synthesize and accumulate some proteins [63].

This result can be explained by different mechanisms that mediate gene silencing or impedes mRNA translation. Thus, several epigenetic marks keep the chromatin condensed, thus impeding gene transcription. Epigenetic processes mediating chromatin structure have been implicated in seed development, and repressed chromatin states through histone posttranslational modifications are responsible for abortion, small size, decreased set, and other abnormal seed phenotypes [64]. Small non-coding RNAs, including short interfering (siRNA) and micro (miRNA) regulate seed development and germination through mRNA cleavage and inactivation [65].

The data reveal that the main differences occurred in the embryo axis. In this direction, proteins related to translation, legumins, proteases, proteasome, and stress-related ones were less abundant in non-germinating seeds. Although there are exceptions to this rule (e.g., fermentative alcohol and lactic dehydrogenase, some ATP synthase, methylglyoxal metabolizing enzymes), the content of enzyme proteins in the different pathways (glycolysis, carbohydrate, amino acid, and lipid metabolism) was lower in non-germinating cotyledons and the embryo axis.

The functional groups and metabolic pathways identified in cotyledons and in the embryo were quite similar. There were some differences in the central, energy, and methyl glyoxal pathways, revealing the particularities of each tissue. Thus, the presence of some proteins was only detected in the embryo axis (alcohol and lactic DH, some enzymes of the TCA) or in the cotyledons (enzymes of the Pentose-phosphate pathway). The set of variable proteins identified in cotyledons and in the embryo axis was different for the carbohydrate, and amino acid pathways. Cotyledons were enriched in reserve proteins and protein-degrading enzymes, clearly pointing to the function of this seed part as a reservoir of nutrients. On the other hand, the embryo axis was enriched in cell defense and rescue proteins, including HSP and antioxidants. All these differences reflect the particularities of each group. The singularity of each seed part, with similar conclusions, has been reported for other plant systems, using transcriptomics and other approaches, including cotton [66] and vetch [67].

To summarize, based on the qualitative differences between some of the variable spots we can propose the proteins within them as biomarkers of seed viability, stage, and tissue. As biomarkers of seed germination, we found that Osmotin proteins were abundant in germinating acorns and absent along the maturation stage. Dehydrin, alcohol dehydrogenase, lactate/malate dehydrogenase and enoyl-ACP reductase precursor were only detected in viable seeds, so they can be considered as biomarkers of seed viability. As markers of storage tissue, we found the legumin precursor and 11S globulin, which were highly abundant in cotyledons. Beta glucosidase and ATP synthase, however, were highly abundant in the embryonic axis and could be considered as biomarkers of the embryonic axis. As markers for stage development, we found that transketolase was highly abundant at middle maturation stages, while dehydrin was highly abundant only at late maturation stages.

From a biological and metabolically point of view, both cotyledons and the embryo axis are characterized by an active metabolism throughout the filling and maturation phases that occur even at the mature acorn stage, with seed development and germination being a continuous process. According to the protein abundance values, we can even conclude that there is a greater metabolic activity in the middle stages of maturation in cotyledon (M6–M7), then moving total ones (M8–M9) in the embryo, considering it as being partly independent. Most of the proteins decreased in cotyledon at M9 and in the germination phase, while the opposite happened in the embryo axis. In cotyledon, most of the energy is assumed to be driven towards the synthesis of reserve nutrients, and, once accumulated, it ceases, moving to the degradation and mobilization of reserves. In the embryo axis, the energy is driven towards synthesis, growth and development [68].

Orthodox and recalcitrant seeds have common characteristics in terms of their changes in gene expression, protein profiles, and metabolism, so seed maturation and germination processes are similar. Just as an example, fermentation is the major pathway supporting ATP and energy demand for reserve synthesis, this being determined by the hypoxic conditions due to low oxygen uptake. Even RubisCO large subunit has been detected in both cotyledons and embryo, although it does not implicate active photosynthesis. Both orthodox and recalcitrant seeds accumulate reserve, antioxidant, and stress-related proteins, in either cotyledon or in the embryo axis.

The main difference between orthodox and recalcitrant seeds is the existence in the former but not in the latter of a desiccation stage associated with the acquisition of drought tolerance, a bridge, as defined by [12], between the maturation and germination phases. This transition and the drought tolerance implicated imply changes in gene expression and metabolic switches that have not been observed in the present proteomics study with *Q. ilex* [69]. Recalcitrant seeds are thus being described as those shedding with an active metabolism, which is supported by the data presented in this work, while orthodox ones maintain almost no metabolic activity [70]. Thus, proteins associated with glycolysis, TCA cycle, cell wall metabolism, are present at high levels at the maturation stage, while associated genes are down-regulated during seed desiccation, and then up-regulated during germination, in orthodox seeds [12,71]. As pointed out by [5],high metabolic rates at the maturing stage is a signature of recalcitrant seeds that lack the quiescent stage at the end of seed development, which characterizes orthodox ones.

However, the difference is never qualitative or clear as black and white. Thus, genes encoding proteins of the translation machinery, DNA repair, proteolysis, and energy metabolism, are highly abundant in the stored mRNA population, and they serve for germination [68]. Despite the absence of proteins, as is the case for HSPs, transcripts are accumulated in the late maturation stages [7].

Desiccation tolerance acquisition has been associated with the accumulation of osmolytes, disaccharides and oligosaccharides, storage proteins, late embryogenesis abundant proteins and HSP, and antioxidative defenses. Out of them, LEAs, and the synthesis of osmolytes, and di- and oligosaccharides have not been visualized in the present proteomics analysis, but they are a general characteristic of orthodox seeds [7], so it is possible to speculate on these absent proteins in *Q. ilex* as being one of the causes of recalcitrance. The presence of the galactinol synthase transcript, implicated in the synthesis raffinose, related to desiccation tolerance, has been noted in *Q. ilex* seeds, with its abundance being maximum at M9 and then decaying at later germination stages [14]. In this direction, it has been suggested that the lack of desiccation tolerance of *A. marina* seeds might be related to the absence of desiccation-related LEAs [72], a fact also reported for other recalcitrant species [73]. LEA transcripts have been identified in *Q. ilex* leaves when they have been induced in response to drought. Other processes could be responsible for their loss of viability, such as the overproduction of ROS, and the decrease in antioxidant defenses [74].

However, the process is complex and there is no simple answer. Aspects such as hormonal balance, and signal transduction pathways, leading to structural and metabolic gene up-regulation have to play a key role; however, this information has remained elusive to the proteomics platform employed in this work so we had to shift to other proteomics and other -omics strategies. Thus, by using GC–MS metabolomics, low levels of abscisic acid and high levels of gibberellins were detected in mature *Q. ilex* acorns [14], which favor seed germination [75].

Finally, epigenetic processes and epigenetic marks mediating gene expression can be, to some extent, responsible for the differences between stages, tissues, viable and non-viable, recalcitrant and orthodox seeds. Epigenetics in orthodox seeds has been studied to some extent [6,76] but has just started to be investigated in recalcitrant ones [77,78].

## 4. Material and Methods

### 4.1. Plant Material

Acorns were collected from Holm oak trees at the Cerro Muriano (Córdoba, Spain; 38°0′0″ north, 4°46′0″ west) location. They were sampled at nine developmental stages throughout a six-month period (June to November 2014), named M1 to M9, according to days after full bloom (DAB) (Appendix A), with M9 corresponding to mature fruits. Only healthy acorns were picked, taken to the laboratory, abundantly washed with tap water, and blot dried with filter paper. Maximum diameter, length and weight were determined, selecting for ulterior analysis those that had similar morphometric parameters (Table 1, Appendix A).

Based on morphometric parameters and water content, the different stages were grouped as follows: early (M1–M3), middle (M4–M7), late (M8–M9) maturation, and early (G1–G3) and late (G4–G5) germination. They corresponded to the Phases I/II (cell division and expansion), III (dry matter accumulation), and IV (water content reduction, desiccation for orthodox seeds). A similar classification has been established previously [79].

Seed parts (cotyledons and embryo axis) were macerated in a mortar under liquid nitrogen until a fine powder was obtained. Powder was stored at −70 °C until protein extraction. Acorns from three different trees, in a number of five (M9) to fifty (M1), depending on the developmental stages, were processed, each tree being considered as a replicate.

Seeds were germinated on humid perlite (50 seeds/replicate, three replicates) in a plant growth chamber under 12/12 h photoperiod (25/20 °C) for two weeks (Appendix A). The perlite was humidified every three days. A 70% percentage of germination was obtained and, as corresponds to non-orthodox species, no total synchronized germination was observed. Only synchronized seeds were sampled at different times, depending on the length of the radicle, corresponding to G1 (2 days, radicle protrusion), G2 (3 days, 1 cm), G3 (4 days, 1.5 cm), G4 (7 days, 3 cm) and G5 (10 days, 5 cm) (Appendix A). As an additional sample, embedded acorns that did not germinate in the two weeks were included (NG seeds).

Relative water content was determined for all the tissues at the different developmental stages, from maturation to germination [14].

### 4.2. Protein Extraction and Gel Electrophoresis

Three independent extractions (biological replicates) for each seed tissue (cotyledon and embryo axes, 0.5 g each) at maturation M4 to M9, and germination G1 to G5 stages were performed. Tissue powder was extracted using the TCA-acetone-phenol protocol [80,81]. The final pellet was dissolved in 40–200 μL of 7 M urea, 2 M Thiourea, 4% CHAPS, Triton X100 2% and 100 mM Dithiothreitol (DTT) (Bio-Lyte 3/10, Bio-Rad, #1631113). Once the pellet was solubilized and the insoluble material eliminated by centrifugation, the protein content was quantified by the method of [82] using bovine serum albumin as a standard.

Proteins (60 μg per replicate) were separated by one-dimensional SDS–PAGE according to [83] and stained with Coomassie Brilliant Blue–R250 [84]. The molecular weight was determined relative to protein markers (SDS–PAGE Standards, Bio–Rad, 161–0304). Band intensity was analyzed by the Quantity-one software (Bio-Rad, Hercules, CA, USA). Normalized band intensity values based on total intensity bands were calculated for each sample line and used for the statistical evaluation of differential abundance.

Two-dimensional electrophoresis was performed in the 5–8 pH range, as this was where most of the proteins were focused [15]. Immobilized pH gradient (IPG) strips (17 cm, 5–8 pH linear gradient; Bio–Rad, #1632004 ) were passively rehydrated for 16 h with 250 μg protein in 300 μLof IEF solubilization buffer (7 M urea; 2 M Thiourea; 4% [*w*/*v*] CHAPS; 0.2% [*v*/*v*] IPG buffer 5–8, 100 mM DTT; and 0.01% [*w*/*v*] bromophenol blue). The strips were loaded onto a Bio–Rad Protean IEF Cell system (Bio-Rad, Hercules, CA, USA) and proteins electro-focused [80]. Strips were equilibrated, subjected to second dimension SDS–PAGE and gels Sypro Ruby (SYPRO^®^Ruby Protein Stains, Bio-Rad, Hercules, CA, USA) stained [85]. Images were analyzed with the PDQuest^TM^ software (Bio-Rad, Hercules, CA, USA), using a guided protein spot detection method [86]. Normalized spot volumes based on total quantity in valid spots were calculated for each 2–DE gel and used for the statistical evaluation of differential protein abundance. Experimental *M*_r_ values were calculated by mobility comparisons with low-range molecular weight standards (Bio-Rad, Hercules, CA, USA ) run in a separate marker lane on the SDS–gel, while p*I* was determined by using a 5–8 linear scale over the total dimension of the IPG strips.

### 4.3. Mass Spectrometry Analysis and Protein Identification

Spots were automatically excised (Investigator ProPic, Genomic Solutions), transferred to Multi well 96 plates and digested with modified porcine trypsin (sequencing grade; Promega, Madison, WI 53711-5399, USA) by using a ProGest (Genomics Solution) digestion station. The digestion protocol used was that of [87] with minor variations. Gel plugs were distained by incubation (twice for 30 min) with 200 mM ammonium bicarbonate in 40% acetonitrile (ACN) at 37 °C, then subjected to three consecutive dehydration/rehydration cycles with pure ACN and 25 mM ammonium bicarbonate in 40% ACN, respectively, and finally dried at room temperature for 10 min. Trypsin (20 µL) at a final concentration of 12.5 ng/µL in 25 mM ammonium bicarbonate was added to the dry gel pieces and the digestion proceeded at 37 °C overnight.

Peptides were extracted from gel plugs by adding 10 µL of 1% TFA (15 min incubation), and then purified by ZipTip. Peptides were deposited onto a MALDI plate using the dry droplet method (ProMS, Genomic Solutions) and the α–cyano hydroxycinnamic acid as a matrix at 5 µg/µL concentration in 70% ACN, 0.1% TFA. Samples were analyzed in a 4700 Proteomics Analyzer MALDI–TOF/TOF Mass Spectrometer (Applied Biosystems, Canada), in the m/z range of 800–4000, with an accelerating voltage of 20 kV, in reflectron mode and with a delayed extraction set to 120 ns. Spectra were calibrated using the trypsin autolysis peaks at *m*/*z* = 842.509 and *m*/*z* = 2211.104 as internal standards. The three most abundant ions were then subjected to MS/MS analysis, providing information that could be used to determine the peptide sequence.

A combined search (MS plus MS/MS) was performed using GPS Explorer^TM^software v 3.5 (Applied Biosystems, Canada) over NCBInr and homemade *Quercus* protein database [88] using the MASCOT v 1.9 search engine (Matrix Science Ltd., London, UK). The following parameters were allowed: taxonomy restrictions to Viridiplantae, one missed cleavage, 10 ppm mass tolerance in MS and 0.5 Da for MS/MS data, cysteine carbamidomethylation as a fixed modification and methionine oxidation as a variable modification. The confidence in the peptide mass fingerprinting matches (*p* <0.05) was based on the MOWSE score and confirmed by the accurate overlapping of the matched peptides with the major peaks of the mass spectrum.

### 4.4. Statistical Analysis

For the statistical and cluster analyses of protein abundance values, the web-based software National Institute of Aging (NIA) array analysis tool was utilized [89]. This software tool selects statistically valid protein spots based on the analysis of variance (ANOVA). After uploading the data table and indication of biological replications, the data were analyzed statistically using the following settings: error model max (average, actual), 0.01 proportions of highest variance values to be removed before variance averaging, 10 degrees of freedom for the Bayesian error model, 0.05 FDR threshold, and zero permutations.

The entire data set was analyzed by principal component analysis (PCA) using the following settings: covariance matrix type, four principal components, 2–fold change threshold for clusters, and 0.5 correlation threshold for clusters. PCA results were represented as a biplot, with consistent proteins in those experimental situations located in the same area of the graph.

To analyze whether the proteins were ubiquitously abundant among the different tissue seed parts, a Venn diagram was plotted by using Venny (http://bioinfogp.cnb.csic.es/tools/venny/index.html). The proteins identified were subjected to heat mapping using Pearson’s distance [90] with the average linkage algorithm of the Genesis software package allowing a definition of some expression groups; the presence or absence of some of them is a characteristic of a particular group [91].

## Figures and Tables

**Figure 1 ijms-21-04870-f001:**
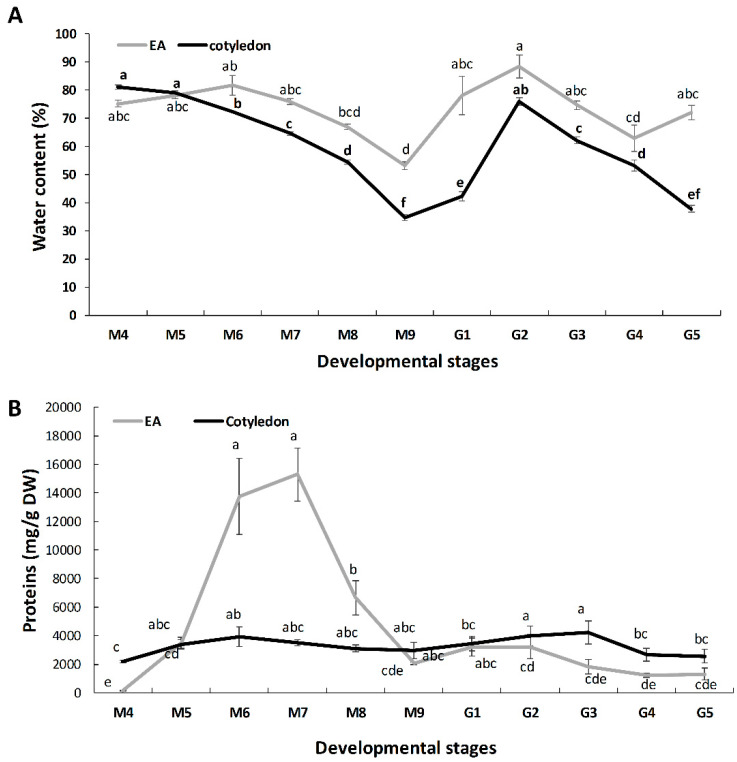
Water (**A**) and protein content (**B**) in seed parts, cotyledon and embryo axis (EA)/radicle, at the different stages of maturation (M1–M9) and germination (G1–G5) processes. Values correspond to the mean of six replicates, with bars corresponding to the Standard error. Different letters indicate significant difference with *p* < 0.05 (Tukey test).

**Figure 2 ijms-21-04870-f002:**
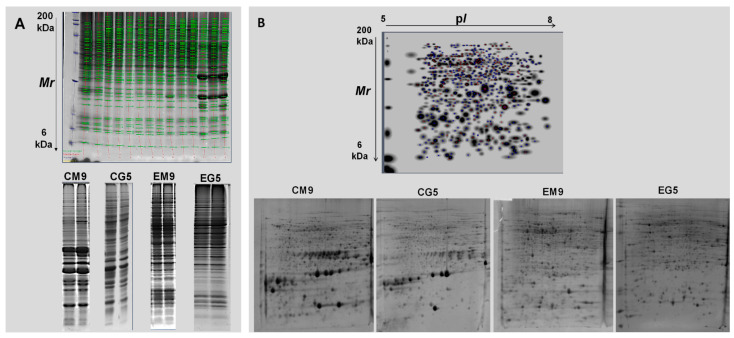
Protein profiles of seed parts, cotyledons and embryo, as visualized by (**A**): 1-D (60 µg of proteins were separated, molecular marker (Mr) is given on the left) and by (**B**): 2-D gel electrophoresis (250 µg of proteins were separated in the first dimension on an immobilized, linear, 5–8 pH gradient and in the second dimension on a 12% acrylamide-SDS gel). Sypro Ruby (SYPRO Ruby Protein Stains Bio-Rad) staining was used to visualize gels following the indications of the instruction manual. Representative images, corresponding to specific stages (M9 and G5) were included. For 2-DE, the master gel is presented. C: cotyledon; E: embryo axis.

**Figure 3 ijms-21-04870-f003:**
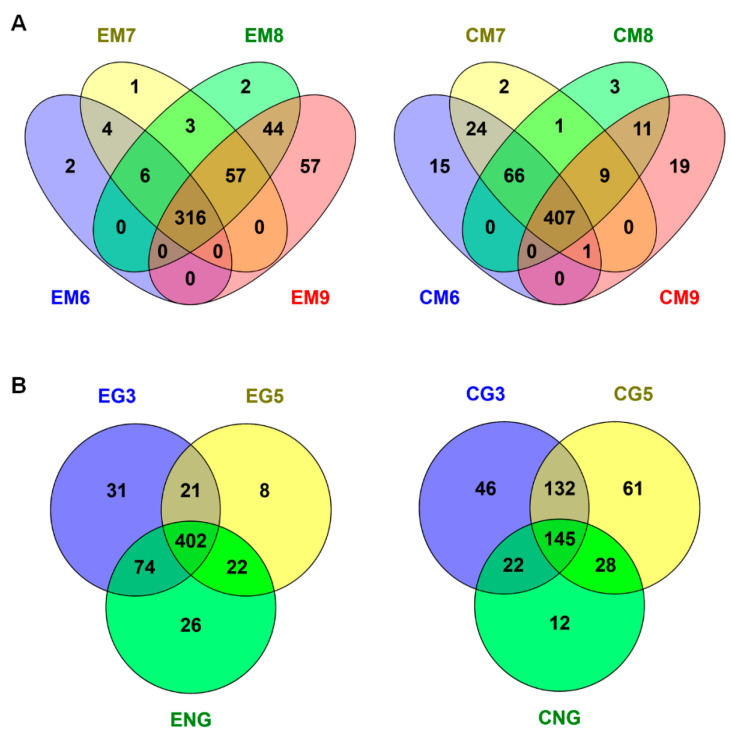
Venn diagrams represent the number of protein spots generated from the 2-DE analyses in cotyledon and embryonic axis during maturation (**A**): from M6 to M9, and during germination (**B**): G3 and G5. The diagrams were plotted using Venny software. The overlapping region between any two groups represents the number of common protein spots between development stages and/or tissues. C: cotyledon; E: embryo; M: maturation; G: germination.

**Figure 4 ijms-21-04870-f004:**
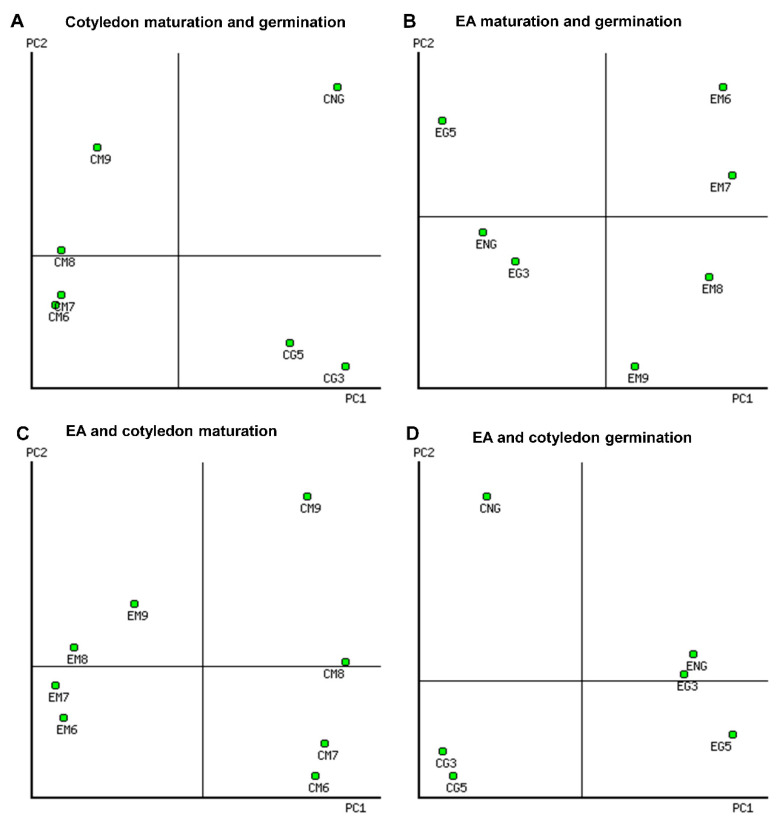
Principal component analysis of 2-DE consistent spots. The plots correspond to cotyledons (**A**), the embryo axis (**B**), maturation (**C**), and germination (**D**). More details are included in Appendix A.

**Figure 5 ijms-21-04870-f005:**
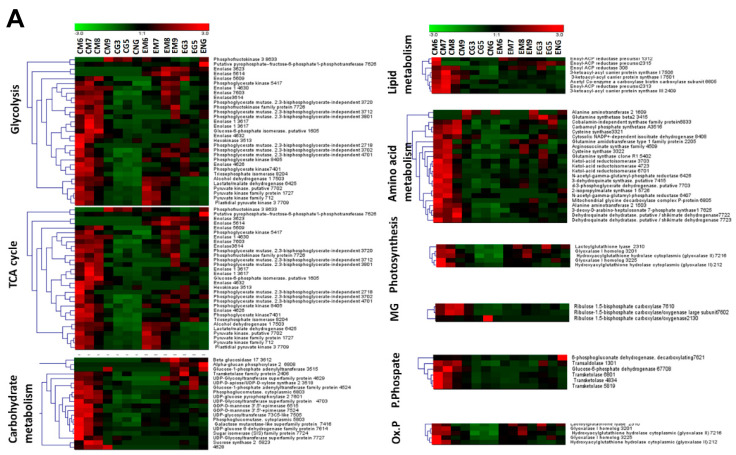
(**A**,**B**) Two-way hierarchical cluster of differentially variable proteins showing changes in abundance in cotyledon and embryo axis along the maturation and germination processes according to the Genesis software package (Eisen et al. 1998). A heatmap representation of the clustered spots shows the protein values according to the level of normalized experiments, which are indicated from −3 (minimum positive expression: clear color) to 3 (maximum positive expression: dark color); black indicates zero expression. Proteins were classified according to the Kyoto Encyclopedia of Genes and Genomes (KEGG).

**Figure 6 ijms-21-04870-f006:**
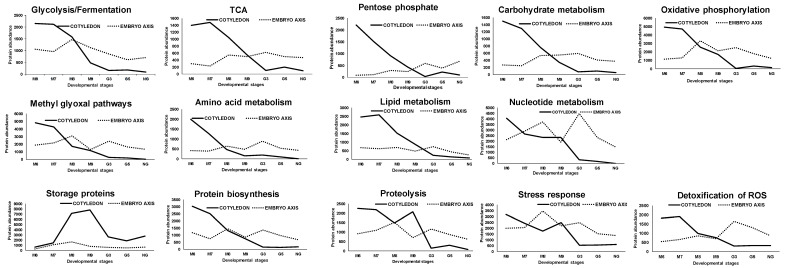
Time course trends of the different protein groups along the maturation and germination stages. Values correspond to the mean abundance of all the proteins within each group (see Figure 5 and Appendix A). Data from non-germinating, unviable seeds have been also included. Black line: cotyledon; point rond line: embryo axis.

**Table 1 ijms-21-04870-t001:** Morphometric parameters (length, maximum diameter) and weight of the acorns at the different maturation stages (M1 to M9). M9 corresponds to the mature fruit. The data are presented as the mean ± Standard error of six replicates. Different letters indicate a significant difference with *p* < 0.05 (Tukey test).

Stages	Length (mm)	Maximum Diameter (mm)	Weight (g)
M1	5.00 ± 0.28f	3.07 ± 0.16g	0.03 ± 0.0003e
M2	5.03 ± 0.24f	3.55 ± 0.32g	0.05 ± 0.0003de
M3	8.61 ± 0.36ef	7.16 ± 0.34f	0.19 ± 0.04de
M4	10.88 ± 0.67de	8.95 ± 0.31e	0.40 ± 0.06de
M5	14.28 ± 0.78cd	10.65 ± 0.39d	0.83 ± 0.068cd
M6	18.04 ± 1.04cd	11.86 ± 0.42cd	1.33 ± 0.11cd
M7	25.73 ± 1.56b	13.42 ± 0.53bc	2.72 ± 0.35b
M8	36.16 ± 1.48a	15.77 ± 0.41a	5.24 ± 0.17a
M9	33.12 ± 0.88a	14.95 ± 1.20ab	3.50 ± 0.30b

**Table 2 ijms-21-04870-t002:** Total number of consistent spots matched between the cotyledon and embryo axis during maturation (M6 to M9) and germination (G3 and to G5) stages. DP: differential protein spots during maturation and germination. NG: non-germinated acorns. The data are presented as the mean ± standard error of three replicates. Different letters indicate significantly difference with *p* < 0.05 (Tukey test).

Matched Spots	M6	M7	M8	M9	G3	G5	NG	DP
Cotyledon	504 ± 4.3a	506.6 ± 1.5a	489 ± 1ab	456.6 ± 23b	248.6 ± 46c	230.6 ± 17c	234.6 ± 23c	558
Embryo axis	325.3 ± 0.53c	382 ± 2bc	422.6 ± 2.8b	470 ± 1a	385.3 ± 25b	243 ± 53d	388.3 ± 62b	591

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
