# Peer review of "Dissecting the Seed Maturation and Germination Processes in the Non-Orthodox Quercus ilex Species Based on Protein Signatures as Revealed by 2-DE Coupled to MALDI-TOF/TOF Proteomics Strategy"

_ijms, 2020, doi:10.3390/ijms21144870_

Round 1

Reviewer 1 Report

The authors made sufficient improvement of the text. Paper can be accepted. 

Author Response

Thanks for your positive response

Reviewer 2 Report

The manuscript is well written and shows a lot of useful data. The manuscript can be accepted for publication after addressing the following comments - 

1. The authors say that the changes in protein profiles were analyzed throughout different acorn developmental and germination stages. However, data are presented only for the middle and late stages. Any explanation?

2. The figure legends need to be expanded for figures 2 and 5.

3. The authors state that "some of the spots showed qualitative differences (presence/absence) between samples and can be
proposed as markers of viability, maturation and germination stages, and tissues ". Did the authors find any other markers? The markers proposed in this research represent what percentage of total markers found by the mass spec?

Author Response

Response to referee 2

Thanks for the positive and valuable comments and criticisms, which have been taken into consideration in the revised version

Comment 1

The authors say that the changes in protein profiles were analyzed throughout different acorn developmental and germination stages. However, data are presented only for the middle and late stages. Any explanation?

Response

Protein profiles of all developmental and germination stages (early, middle and late) were analyzed using the 1 DE approach. Based on the obtained results and statistical analysis, most changes in proteins profiles were observed in middle and late stages. For that, we focused only on those stages to perform the 2DE and identify proteins via MALDI/TOFF_TOFF.

Comment 2

The figure legends need to be expanded for figures 2 and 5.

Response

Figure legends were expanded as follow;

Figure 2. Protein profiles of seed parts, cotyledons and embryo, as visualized by A: 1-D (60 µg of proteins were separated, molecular marker (Mr) is given on the left) and by B:2-D gel electrophoresis (250 µg of proteins were separated in the first dimension on an immobilized, linear, 5–8 pH gradient and in the second dimension on a 12 % acrylamide-SDS gel). Sypro Ruby (SYPRO Ruby Protein Stains Bio-Rad) staining was used to visualize gels following the indications of the instruction manual. Representative images, corresponding to specific stages (M9 and G5) were included. For 2-DE, the master gel is presented. C: cotyledon; E: embryo axis.

Figure 5. Two-way hierarchical cluster of differentially variable proteins showing changes in abundance in cotyledon and embryo axis along the maturation and germination processes according to the Genesis software package (Eisen et al. 1998). A heatmap representation of the clustered spots shows the protein values according to the level of normalized experiments which are indicated from -3 (minimum positive expression: clear color) to 3 (maximum positive expression: dark color); black indicates zero expression. Proteins were classified according KEGG.

Comment 3

The authors state that "some of the spots showed qualitative differences (presence/absence) between samples and can be
proposed as markers of viability, maturation and germination stages, and tissues ". Did the authors find any other markers? The markers proposed in this research represent what percentage of total markers found by the mass spec?

Response

No more markers were found. We found 31 identified qualitative differential proteins which are specific to viability, tissues or developmental stages. These biomarkers represent about 10% of identified proteins (in total we have identified 321 proteins). This information was being added in the Ms, page 10 lane 195.

This manuscript is a resubmission of an earlier submission. The following is a list of the peer review reports and author responses from that submission.

Round 1

Reviewer 1 Report

It will be nice to insert line numbers through whole manuscript, not page by page.

It is interesting paper about mechanistical aspect of differences between germinated and non- germinates seeds of Holm oak. Authors performed precise analysis of protein abundance at different stage of development and it characterization.

There are some points need to be clarified and corrected.

However, authors did not take into account epigenetic aspect of seeds maturation and germination, what is the main reason of differences in seeds status. It should be mentioned at least in the introduction and discussion. And should be the subject of future research.   

Abstracts: please, make it more easy, avoid abbreviations and complicated statement.

This part must be clear itself. You mention stages M6-M8; M7-M9, but how reader can now what do you mean without reading the whole text?     

Introduction. One of the most important process during seeds maturation/germination is changes in epigenetic status, but these citations are missing. Please, add it!

Results.

Page 3 line 11: please, provide characterization of developmental stages. You mention M4-M9 and G1-G8, but did not explain.

Page 4, line 1: Please, cut comma: “maximum, diameter“.

Page 4, line11-12: this is from M&M, Please, re-formulate.

Line 17: “values of around 2-3 g/g DW”? 3 g per g weight?

Line 20: “The protein extracts from the different developmental stages (M4 to G5)” – protein can not be extracted form stages, from seeds at different stage.

Page 15: “Our own unpublished results“ ??

Page 16, line 27: “defence and stress-related genes” -do you activation of gene expression?

Page 17: “The simple visualization of the gels showed that the number of spots was much lower in non-viable, non-germinating seeds“- that’s true, but the reason of this differences is., most probably, chromatin condensation what does not allow to activate gene expression. It should be discussed as well.

Literature: please, make better layout! There is a big variation in layout between citation.

Author Response

REFEREE 1

Thanks for the positive and valuable comments and criticisms, which have been taken into consideration in the revised version

Comment 1

It will be nice to insert line numbers through whole manuscript, not page by page.

In our original manuscript submission, the lines were numbered continuously, so we think that the changes in line numbering were done by the journal, according to the writing instructions. The revised version is submitted with a continuous numbering.

Comment 2

It is interesting paper about mechanistical aspect of differences between germinated and non- germinates seeds of Holm oak. Authors performed precise analysis of protein abundance at different stage of development and it characterization.

There are some points need to be clarified and corrected.

However, authors did not take into account epigenetic aspect of seeds maturation and germination, what is the main reason of differences in seeds status. It should be mentioned at least in the introduction and discussion. And should be the subject of future research.

You have mentioned a very important point that we did not consider as we did not perform an epigenetic analysis. Because of that we omitted this important issue. Following your comments, we took the liberty to introduce how epigenetics contribute to the seed maturation and germination program, which is illustrated with a couple of references. A more thorough explanation was given in the discussion section. We fully agree with your statement and we have started to optimize some methodologies to use in analysing epigenetic changes that will help us in the explanation of variability, development and responses to stresses in Quercus. How epigenetic determine gene expression programs was mentioned in the introduction and minimally speculated in the discussion section. We have further included a reference on that.

Comment 3

Abstracts: please, make it more easy, avoid abbreviations and complicated statement.

The abstract has been rewritten following the referee indications.

Comment 4

This part must be clear itself. You mention stages M6-M8; M7-M9, but how reader can now what do you mean without reading the whole text?  

We understand what you mean and fully agree with your comment. The different stages are characterized by morphometric and water content parameters, as indicated in Table 1, Figure 1 and Suppl. Figures S1 and S2. Using MX term favoured the writing of the papers, but probably made its reading and understanding difficult. In order to be self-explanatory, we have categorized the different stages, and based on data in Figure 1 and Table 1, as early (up to M3), middle (M4-M7), and late (M8-M9). Similarly, the germination stages divided in early (G1-G3) and late (G4-G5) stages. Of course it is an arbitrary classification; based on data in Table 1 and Figure 1, as there are no molecular markers defining precisely each stage and we did not analyze hormones. In any case, they corresponded to the phases reported in the literature, cell division and expansion (Phases I and II, corresponding to the denominated early in our manuscript), dry matter accumulation (Phase III, corresponding to the denominated middle in our manuscript), and desiccation (Phase IV, of course for orthodox seeds, corresponding to the denominated late in our manuscript, in which water content is significantly reduced). A similar classification of seed developmental stages has been done[1].

This was indicated in the material and methods section and later on, in results.

Comment 5

Introduction. One of the most important process during seeds maturation/germination is changes in epigenetic status, but these citations are missing. Please, add it!

It has been commented in the introduction as you indicate. We have included a couple of references, no more, as referee 2 ask us to shorten the cited literature.

Comment 6

Page 3 line 11: please, provide characterization of developmental stages. You mention M4-M9 and G1-G8, but did not explain.

This point has been answered (see comment 4).

Comment 7

Page 4, line 1: Please, cut comma: “maximum, diameter“.

Done, thanks

Comment 8

Page 4, line11-12: this is from M&M, Please, re-formulate.????????

Done, thanks

Comment 9

Line 17: “values of around 2-3 g/g DW”? 3 g per g ?

Corrected, thanks

Comment 10

Line 20: “The protein extracts from the different developmental stages (M4 to G5)” – protein cannot be extracted form stages, from seeds at different stage.

Corrected, thanks

Comment 11

Page 15: “Our own unpublished results“ ??

The data are part of a manuscript which is now being evaluated. To avoid confusion, the sentence has been eliminated.

Comment 12

Page 16, line 27: “defence and stress-related genes” -do you means activation of gene expression?

Corrected, thanks

Comment 13

Page 17: “The simple visualization of the gels showed that the number of spots was much lower in non-viable, non-germinating seeds“- that’s true, but the reason of this differences is., most probably, chromatin condensation what does not allow to activate gene expression. It should be discussed as well.

Thanks a lot. This hypothesis has been now incorporated in the revised manuscript within the discussion section, it accompanied by a reference.

[1]Similarity between soybean and Arabidopsis seed methylomes and loss of non-CG methylation does not affect seed development

Jer-Young Lin, Brandon H. Le, Min Chen, Kelli F. Henry, JungimHur, Tzung-Fu Hsieh,  View ORCID ProfilePao-Yang Chen, Julie M. Pelletier, Matteo Pellegrini, Robert L. Fischer, John J. Harada, and Robert B. Goldberg

PNAS November 7, 2017 114 (45) E9730-E9739; first published October 23, 2017 https://doi.org/10.1073/pnas.1716758114

Reviewer 2 Report

The MS presents some interesting results, but it needs a major revision because the data concern mainly on the number of different protein spots (Table 2, Figure 3-4) something that is meaningless because related to the size of the gels.

So, Authors must rewrite the MS describing and discussing the meaning and relevance of the 178 protein spots identified citing some of them which could have a relevant meaning during germination.

Moreover:

  1. Authors must reduce the reference number; in Introduction, in less than 50 lines are listed 40 references!
  2. Line 64: spots after a SDS-Page are not proteins;
  3. 1-D gel electrophoresis is meaningless and whereas 40-50 bands resolved is a poor result;
  4. Table 3 is incomprehensible;
  5. Rubisco as itself cannot be present after SDS-PAGE, Authors must refer to small and large Rubisco subunits.

Author Response

REFEREE 2

Thanks for the positive and valuable comments and criticisms, which have been taken into consideration in the revised version

Comment 1

The MS presents some interesting results, but it needs a major revision because the data concern mainly on the number of different protein spots (Table 2, Figure 3-4) something that is meaningless because related to the size of the gels.

Thank you so much for your comment.  For sure, 2-DE has limitations; however, it is still a valuable and trusted technique in the literature. In fact, more than 80 % of the plant proteomics papers published are based on this method. It is true that currently the Shotgun technique is more widely used and more potent than 2-DE, but still both techniques are complementary. Further, despite being a powerful technique; Shotgun has some serious limitations such as confident protein quantification and identification. In fact, we are now (if the coronavirus allows us) running experiments to analyse changes in the protein profiles using a shotgun platform.

Comment 2

So, Authors must rewrite the MS describing and discussing the meaning and relevance of the 178 protein spots identified citing some of them which could have a relevant meaning during germination.

In our manuscript, most of the variable proteins and their roles in seeds development and germination were discussed. Indeed, at the end of discussion; we further emphasized at the most relevant proteins and compared them between seed parts, viable and inviable, orthodox and recalcitrant seeds.

Comment 3

Authors must reduce the reference number; in Introduction, in less than 50 lines are listed 40 references!

We perfectly understand your comment, so we reduced the number of references to keep only the most relevant ones in the revised version. However, it was our intention to support our findings with the previously published data and that’s what made our reference list long.

Comment 4

Line 64: spots after a SDS-Page are not proteins;

Thank you so much, it is fully true, and we have corrected it in the revised version.

Comment 5

1-D gel electrophoresis is meaningless and whereas 40-50 bands resolved is a poor result;

Yes, it is true that 1-D gel electrophoresis is not a very powerful technique to analyse significant protein changes. Nevertheless, it is still a very helpful technique that allowed us to discriminate the most different stages along the maturation or germination processes. Those more different were subjected to 2DE analysis. To perform a 2-DE analysis on all stages was not realistic (16 in total, 48 2-DE gels counting the replicates). It is clearly indicated in the manuscript itself.

Comment 6

Table 3 is incomprehensible;

We are very sorry about it. It was not our intention. This table was a key for us while writing the manuscript so we expected it would help in reading and understanding it. There are so many data coming from omics approaches, and a number of stages in the present manuscript so it was necessary to define trends of time evolution for the different groups. This table was the best solution we could find to summarize the results. In any case, it has been modified, so we hope it is clearer now.

Comment 7

Rubisco as itself cannot be present after SDS-PAGE, Authors must refer to small and large Rubisco subunits.

You are right. In our hands with quite a number of plant systems we only detect the large subunit of the RubisCO. For whatever reason, the small one is not visualized. For sure it must be present. We extrapolated the RubisCO LS data to the whole enzyme. The text has been modified accordingly and we hope now it is clearer in the revised version.

Round 2

Reviewer 2 Report

I am sorry, but the authors' response was not satisfactory, and the manuscript retains the problems reported in the previous revision; therefore, it still needs a major revision.

In fact, I did not criticize the methodology but the fact that the results are represented by the number of spots (and also by spots after a simple SDS-PAGE: 1-D gel electrophoresis is meaningless as the resolution of 40-50 bands). In fact, Table 2 and Figure 3 and 4 they all concern the number of spots, and do not give any information about specific or unique proteins present at a particular stage of development.

Furthermore, the Authors still confuse spots with proteins whereas spots represent only polypeptides or part of polypeptides (e.g. lines 170, 204-206).

Figure 5 should be divided in two or three, otherwise it's illegible.

Table 3 is still incomprehensible /unreadable.

The discussion about identified proteins appears as a mere list with little connection to the germination process or the identification of markers.

The protein level was practically constant in cotyledons, but it changed considerably in embryo axis (Figure 1B); therefore, it seems strange the lack of increase in proteolytic enzymes in embryo axis from stage M6 to M9 (Table 3).

Author Response

Regarding the manuscript which is under evaluation we are in a dead point and honestly, we do not know how to leave this situation.

Referee comment

In fact, I did not criticize the methodology but the fact that the results are represented by the number of spots (and also by spots after a simple SDS-PAGE: 1-D gel electrophoresis is meaningless as the resolution of 40-50 bands).

Our answer

Our work and manuscript fits in the standards and MIAPES requested for a 2-DE based plant proteomics publication. There is not a critic to the methodology, so I assume he/she considers it valid to study plant biological processes. For comparative purposes, 2-DE proteomics is based on differences in spot intensity which is statistically analysed to identify variable spots. Once you identify proteins based on MS data you can infer differences in protein species abundance. It is the number of spots and their intensity that determines differences among samples. It is after MS that you put the name to the proteins within the variable spots.

We have clearly explained in the R1 manuscript and in the response to the reviewer that 1-DE was used to stablish which samples were the most different, that is all. We had 14 samples, three replicates each which is impossible to handle in a 2-DE strategy (42 gels). The number of samples were reduced for later 2-DE analysis based on the band pattern, that is all. It is obvious that 1-DE has poor resolution but was useful for discriminating and grouping samples.  For example, have a look to Jorrin Novo, Komatsu et al. 2019. Gel electrophoresis-based plant proteomics: Past, present, and future. Happy 10th anniversary Journal of Proteomics! Journal of Proteomics 198, 1-10

Referee comment

In fact, Table 2, and Figure 3 and 4 they all concern the number of spots, and do not give any information about specific or unique proteins present at a particular stage of development.

Our answer

What else? Have a look to the current plant proteomics literature in which this approach has been used. All of them analyse the spots profile and the spot intensity, perform a statistical analysis, reveal which are variable, according to statistical tests, and finally identify the proteins within the spots. Do you want we delete all these data and figures? If so, please, you should said us. I insist, this is how a 2-DE plant proteomics paper is and should be presented.

The second sentence is not true. Have a look to the discussion section, in which we dealt with almost all the variable proteins and the functional groups they belong to. To be honest, this is an opinion by the referee, which is not justified. Referee 1 did not have the same opinion. You should not expect unique proteins in samples, which is quite uncommon, but differences in protein abundance inferred from spot intensity.

Referee comment

Furthermore, the Authors still confuse spots with proteins whereas spots represent only polypeptides or part of polypeptides (e.g. lines 170, 204-206).

Our answer

If the referee observation is right, I am more stupid than ever imagined. Please, check my CV at PubMed, WoS, Research Gate. How many papers I have published, reviewed, edited? How many reviews I have performed, and how many cites they have? If I confuse spots and proteins all the work I have done is just rubbish. I think I have got some confidence within the plant proteomics community, or at least for your journals, from which I have received invitations several times. The last to co-edit with Prof. Komatsu the Plant proteomics 3.0 issue to which the manuscript has been submitted

Anyway, the answer: being strict, before MS we must talk about spots, and after MS of protein, or better proteoforms or protein species. It is true that both terms are interchangeable and used as synonymous, and it can generate some confusion.

Referee comment

Figure 5 should be divided in two or three, otherwise it is illegible.

Our answer

This uses to happen with this type of figures (check current literature). Even me do not like it very much, but this is the standard for the -omics era. Anyway, there are alternatives for an on line journal to manage its view.

Referee comment

Table 3 is still incomprehensible /unreadable.

Our answer

We have got a different opinion, and we did not have any comment from referee 1 in this regard. Anyway, let assume the referee is right. What does she/he want? Delete, change, or what? I think the evaluation comments are sometimes merely opinions as there is not a clear explanation or even a suggestion of how to modify it. This is what referee 1 did. In any case, we have changed it to Figure 6, showing the evolution trend for the different functional groups.

Referee comment

The discussion about identified proteins appears as a mere list with little connection to the germination process or the identification of markers.

Our answer

Once again, this is an opinion which is not true. Proteomics gives us list of proteins that are translated to biology. Proteomics is a descriptive discipline, and it is obvious, and it is what we have done. Based on spot intensity and variations and from the list of identified proteins we have proposed:

  1. Which the differences between viable and non-viable seeds are.
  2. Which the differences between cotyledon and embryo are.
  • Which the differences along the seed maturation and germination are.
  1. We have hypothesized on the differences between orthodox and recalcitrant are.

To expect clear markers from this type of work is ignoring the field (plant proteomics). In any case, we have discussed or better speculated about the difference in between orthodox and recalcitrant seeds, and also between viable and non-viable seeds. Please, read the manuscript carefully and focus on the LEA and its absence been associated to recalcitrance, which has also been previously suggested.

Referee comment

The protein level was practically constant in cotyledons, but it changed considerably in embryo axis (Figure 1B); therefore, it seems strange the lack of increase in proteolytic enzymes in embryo axis from stage M6 to M9 (Table 3).

Our answer

It is true, but it is from M7 to M9. We have written with respect to the proteases in the manuscript since the original submission:

Maximum values in the embryo axis took place at M8, keeping it constant at germination stages, except in non-germinating seeds.

So, the answer is given in the manuscript itself.

Round 3

Reviewer 2 Report

The Author's approach is completely anti-scientific. The criticism and opinions of a reviewer concern the state of the work, not the researcher. Furthermore, the review process is based on the possibility of criticism to which the Author should respond with at least logical arguments, not with a simply “I am right because I am a good researcher”.

I really don't think that an excellent researcher has the right to publish a work just because he is so good and collaborates with equally good people. The last pearl is that opinions can be false, opinions remain opinions and have very little to do with what is true.

So, specifically:

Referee comment

In fact, I did not criticize the methodology but the fact that the results are represented by the number of spots (and also by spots after a simple SDS-PAGE: 1-D gel electrophoresis is meaningless as the resolution of 40-50 bands).

Authors should only to avoid claiming a SDS-PAGE which has separated 40-50 bands (the 1D was the problem not 2D) whereas the samples’ choice should be explained in terms of the importance of the stage of development. Anyway, it is ok now.

Referee comment

In fact, Table 2, and Figure 3 and 4 they all concern the number of spots, and do not give any information about specific or unique proteins present at a particular stage of development.

What else? The Authors could simply produce the same figures using the polypeptides (proteins) identified, this is certainly much more interesting and that will aid to discuss the differences in the identified protein abundance inferred from spot intensity.

Furthermore, the Authors still confuse spots with proteins whereas spots represent only polypeptides or part of polypeptides (e.g. lines 170, 204-206).

I repeat that I don't have any personal problem with the authors I don't even know, but surely each spot corresponds to a polypeptide or part of a polypeptide and not a protein. This has been pointed out in order to improve the MS. They are only two phrases, nothing to do with the Author cv and Prof. Komatsu.

Referee comment

Figure 5 should be divided in two or three, otherwise it is illegible.

I remain of the idea that Figure 5 would be unreadable, it is simply an opinion that can be ignored or simply considered after acceptance.

Table 3 is still incomprehensible /unreadable.

It is not the task of a reviewer to demand specific mandatory changes (which would perhaps be immediately approved to have the MS accepted) but the task is to induce the Authors to introduce changes of their choice that improve the MS. Anyway, there are many possibilities, for example I would prefer to introduce a color scale as shown in Figure 5. The new Fig. 6 is not a perfect choice.

Referee comment

The discussion about identified proteins appears as a mere list with little connection to the germination process or the identification of markers.

So, it remains unclear how many biomarkers have been identified (see line 199 of the present version)